# Eating Concerns Associated with Nutritional Information Obtained from Social Media among Saudi Young Females: A Cross-Sectional Study

**DOI:** 10.3390/ijerph192416380

**Published:** 2022-12-07

**Authors:** Mona Mohammed Al-Bisher, Hala Hazam Al-Otaibi

**Affiliations:** Department of Food and Nutrition Science, College of Agricultural and Food Science, King Faisal University, Al-Ahsa 31982, Saudi Arabia

**Keywords:** social media, eating concerns, SCOFF, ESP, Saudi females, nutritional information, eating disorders

## Abstract

Eating disorders have been highly prevalent in young females for decades for many reasons. Social media platforms have an enormous impact on users, especially young adults, who use them every day. In Saudi Arabia, social media is popular, with an estimated 72% of users being active in 2020. Therefore, the primary aim of this study was to assess the relationship between using social media to search for nutritional information and eating concerns. A nationwide study was conducted on 1092 young Saudi females aged 18–30 years from five administrative regions in the Kingdom of Saudi Arabia. Data were collected using an online validated questionnaire, and symptoms of eating concerns were assessed using two brief instruments: SCOFF [Sick, Control, One Stone, Fat, Food] and Eating disorders Screen for Primary care [ESP]. The prevalence of eating concerns was 49.6% among Saudi females. Moderate eating concerns were more prevalent in the central region 24.8%, whereas high eating concerns were more prevalent in the southern region 27.6%. Personal accounts of dietitian/nutritionists (OR = 1.170; 95% CI 1.071–1.277; *p* ≤ 0.001), interaction with offered experiments about new meals/restaurants, and diets on social media that were mostly promoted by celebrities/influencers (OR = 1.554; 95% CI 1.402–1.723; *p* ≤ 0.000) were the most prominent risk factors associated with being more likely to suffer from eating concerns. The present study recommends opening clinics specializing in nutrition on social media platforms that target young females to provide nutritional counselling and encourage a healthy lifestyle. In addition, it is important to plan awareness campaigns intended to educate young females on how to deal with messages that circulate on social media without any evidence regarding their truthfulness.

## 1. Introduction

The world today is undergoing a digital transformation and tends to depend on technology to perform tasks easily. Social media [SM] is a notable example. People sensed its importance during the COVID-19 pandemic, and it was the first destination to enable following the latest news and communicating with others, regardless of distances and geographical boundaries. Recently, SM has become an important part of the lifestyles of many [1].

The term “social media” refers to a group of applications established on the internet that enable the creation and sharing of content generated by users, and they are constructed on the technological and ideological foundations of Web 2.0 [2]. It includes six types: blogs and microblogs, content communities, social networking sites, virtual communities, collaborative projects, and virtual game worlds [3]. In Saudi Arabia, active users of SM were estimated to be 72% in 2020 [4]. The number of users for common social media applications reached 14.35 million on Twitter [4], 12 million on Instagram [4], and 16.10 million on Snapchat [4].

Previous research has revealed that young adults tend to adopt SM as a source of information regarding many issues, such as public health, fitness, nutritional supplements, food recipes, and exercise [5,6,7]. Many studies have reported that females are more likely to use SM to search for nutritional information [NI] than males [8,9]. These applications are designed to display visual content, including audio and graphics, which might make information more effective and attractive [10]. Therefore, they may represent a new opportunity for dietitians/nutritionists to provide nutritional counselling and communicate relevant information about food and nutrition [11].

SM is widely used among younger people [12]. Young adults are considered to be in a critical transitional period from adolescence to adulthood, which is associated with many changes, such as rapid growth in the body, in addition to developing new behaviours and practices, for example, dietary habits, which depend on nutritional knowledge [13]. The eating habits of millennials are remarkably different from those of their parents and grandparents [14]. SM might be one of the many factors affecting their eating and drinking, particularly because it is becoming a prominent source of information regarding food choices and recipes [15,16].

Although SM can enhance awareness of healthy nutrition, there is a big challenge to face: according to Rambaree et al. [17], frequent use of SM was associated with health problems (e.g., stomach aches, anxiety), and unhealthy eating habits were a contributing factor to those problems. Moreover, Sidani et al. [18] revealed that using SM for more than two hours daily increases the probability of suffering from eating concerns [EC] (OR = 2.18; 95% CI 1.50–3.17; *p* ≤ 0.001). Some of the possible explanations for this observation are that SM applications are overflowing with pictures and videos. Therefore, excessive exposure might promote accidental appearance comparisons, which are considered a sociocultural factor impacting body image [BI] among females [19], which in turn leads to engagement in disordered eating behaviours [DE], including fasting, dieting, binge-eating, calorie counting, and self-induced vomiting [20].

EC constitutes an early sign of developing and maintaining eating disorders [ED] later, which are classified by the American Psychiatric Association [APA] as psychiatric disorders that most affect females aged 12–35 years. It includes three main types: anorexia nervosa [AN], bulimia nervosa [BN], and binge eating disorders [BED] [21]. Each type can be diagnosed by using narrow and specific criteria. AN is mostly associated with body image disturbance, which leads to seriously low body weight due to the restriction of energy intake [22], in which body mass index [BMI] reaches 17.5 kg/m^2^ and below [23]. BN is characterized by recurrent episodes of binge eating, followed by inappropriate compensatory behaviours, such as misuse of laxatives and self-induced vomiting [24], whereas binge-eating episodes in the BED case occur in the absence of compensatory behaviours that prevent weight gain [25].

Many studies have emphasized that ED is related to increasing health complications; for instance, AN is correlated with elevated mortality rates [26] due to harmful cases of being underweight and malnutrition [27]; some cases of ED can lead to being overweight [28]; and it disrupts social functioning [29]. Therefore, it constitutes a burden on public health if uncontrolled [30]. EDs are prevalent worldwide, especially among females, and their prevalence has doubled in a decade (i.e., from 3.5% during the 2000–2006 period to 7.8% during the 2013–2018 period) on three continents: America, Europe, and Asia [31]. In Saudi Arabia, many females (university age) are at risk for ED: 26.66% in Arar [32], 29.4% in Dammam [33], 35.4% in Taif [34], and 36.8% in Riyadh [35].

Despite Saudi females having pronounced activity on SM, with an estimated percentage use of 50.3%, 37.6%, and 30.8% on Snapchat, Instagram, and Twitter [4], respectively, there is a lack of data on the extent of their use for nutrition education and if the use is related to suffering from EC. Although several studies in Saudi Arabia have explored the potential risk factors associated with ED, including (but not limited to) BMI, body image dissatisfaction, and pressure to lose weight [32,33,34], SM did not get adequate attention. Therefore, the added value of this study lies in its being conducted on a large sample of Saudi females from all regions and across a wide age range extending from the end of adolescence to youth, by utilizing instruments that have not been applied previously in Saudi Arabia to achieve the following objectives:(1)Estimating the prevalence of eating concerns [EC] among Saudi females.(2)Investigating whether an association exists between using social media [SM] as a source of nutritional information [NI] and eating concerns [EC]; and(3)Determining the risk factors for suffering with eating concerns [EC].

## 2. Materials and Methods

### 2.1. Study Design and Sample

This cross-sectional study used a non-probability convenience sampling technique to collect data from five administrative regions in the Kingdom of Saudi Arabia [KSA]. Participation in this study was voluntary and included the following inclusion criteria: Saudi females only, aged between 18–30 years, users of at least one of these applications (Twitter, Instagram, and Snapchat) to search for NI, non-pregnant, and non-breastfeeding. The study excluded those who did not have those criteria. The sample size was calculated using Daniel’s [36] formula, which resulted in a minimum of 310 participants.

### 2.2. Study Instruments

Data were collected using a validated questionnaire distributed through different SM applications. The questionnaire was divided into four parts: the first part included questions about demographic characteristics, such as age, marital status, and region of residence. The second part included two questions about health status (How would you say your health? good, average, and poor) and about body weight and height. Then, the authors calculated BMI and classified it according to World Health Organization guidelines [37].

Searching for nutritional information on social media is the third part, and there were two sections. The first part was about the favourite application for searching for NI, the preferred source for NI published in the Arabic language by Arab speakers, frequency of searching, number of minutes spent on search, the interesting topics, and the favorite source for NI on SM. The second part included two questions about the impact of SM influencers/celebrities on the nutritional behaviours of their followers, most notably, young females. This study focused on three famous applications in Saudi Arabia based on their high usage rates among females [4].

Eating concerns is the last part. Two brief screening tools were used to estimate the prevalence of EC: SCOFF [Sick, Control, One Stone, Fat, Food] [38] and the Eating Disorders Screen for Primary Care [ESP] [39]. It should be noted that these tools have been applied in several previous studies on different samples. SCOFF has been translated into many languages, such as Swedish [40], Spanish [41], and Danish [42].

In the present study, the Arabic version of the SCOFF (five items) was used, which was validated by Aoun et al. [43]. As there is no Arabic version of the ESP (four items), the authors have forward translated from English into Arabic while maintaining the idea of the original scale and using language appropriate to their culture. Two bilingual native English-speaking translators (without nutritional or medical background) backward translated the initial translation to make sure that the Arabic version reflected the same item contents of the original version. The translated versions were reviewed by two health educators to identify the content validity. A focus group was also held with six subjects to assess face validity and ensure comprehension of the items in their intended format.

To evaluate the diagnostic ability of the instruments, sensitivity and specificity were calculated using the receiver operating characteristic curve [ROC] test. Both tools [SCOFF, and ESP] achieved high scores (ROC = 1.0). Moreover, when merging them, they achieved a high score (ROC = 1.0). The ROC curve level of more than 0.90 indicates excellent discrimination power [44].

These tools included dichotomous questions (Yes/No). For SCOFF, every answer with (Yes) was considered abnormal, which takes one score, whereas every answer with (No) takes zero, which was considered normal. Similarly, ESP classified abnormal answers as: (No) to questions 1 and 3 and (Yes) to questions 2, and 4. Authors calculated the total scores for SCOFF and ESP, which ranged from 0 to 9, based on a prior study [18]. The cut-off score was divided into three categories by Sidani et al. [18]: Low Eating Concerns (LEC, 0–2), Moderate Eating Concerns (MEC, 3–6), and High Eating Concerns (HEC, 7–9). Therefore, the authors supposed that dividing the sample into three independent groups would make the results more consistent and impartial, which can help to avoid prejudice diagnosis.

### 2.3. Statistical Analysis

Descriptive statistical tests were performed, including frequencies, percentages, means [*M*], and standard deviations [*SD*]. Advanced statistical tests were performed, including comparison tests for more than two independent groups, such as chi-square for categorical variables and one-way ANOVA for continuous variables, correlation coefficients, and the multinomial logistic regression model.

All statistical analyses were performed using the Statistical Package for the Social Sciences [SPSS] version 23, and *p*-values of ≤0.05 were considered statistically significant.

## 3. Results

A total of 1092 Saudi young females participated in this study, of whom 50.3%, 47%, and 2.7% had LEC, MEC, and HEC, respectively.

### 3.1. Sample Characteristics

Of the study population, 24.8%, 22.7%, 20.3%, 17.1%, and 15% reported they were from the central, eastern, western, northern, and southern regions, respectively. The mean age for the overall sample was 23.02 ± 3.47 years, and over half of the participants were unmarried 83.2% (Table 1).

As shown in Table 1, most participants perceived themselves to be in good health, i.e., 60.2%, while 16.1% were dieting. Furthermore, 56.2% of the participants had a normal BMI, whereas being overweight and obese were more prevalent among participants who suffered from MEC and HEC.

### 3.2. Prevalence of Eating Concerns

#### 3.2.1. SCOFF [Sick, Control, One-Stone, Fat, Food]

Of the 1092 participants, 451 (41.3%) reported 2 or more yes responses (positive SCOFF), which is considered a risk factor for ED development (Table 2).

#### 3.2.2. Eating Disorders Screen for Primary Care

Of the total sample, 484 (44.3%) reported 2 or more yes responses (positive ESP) (Table 2).

### 3.3. Using Social Media for Nutritional Information

Table 3 shows that Instagram was most popular among participants (53.8%), followed by Twitter (38.6%) and Snapchat (7.7%). Thirty-one percent of those who had HEC were searching for NI weekly.

When asked about the most interesting topics in nutrition, the majority of those who had MEC and HEC were more focused on diets such as keto, intermittent fasting, and Atkins (49.1% and 51.7%, respectively) and healthy recipes (57.9% and 58.6%, respectively). Nearly 71% of participants preferred the personal accounts of dietitians/nutritionists as a source for NI.

The participants with HEC were less likely to resist celebrity SM advertisements (34.5%) and were more attracted to their experiments (48.3%) than those with LEC (13.3% and 5.5%, respectively).

### 3.4. Risk Factors for Eating Concerns

Multinomial logistic regression showed many factors that predict EC. The most remarkable were dieting (OR = 1.613; 95% CI 1.464–1778; *p* ≤ 0.000), searching for NI on SM for 31–60 min (OR = 1.090; 95% CI 1.000–1.187; *p* ≤ 0.049), personal accounts of dietitians/nutritionists as a favorite source for NI (OR = 1.170; 95% CI 1.071–1.277; *p* ≤ 0.001), interaction (OR = 1.170; 95% CI 1.072–1.277; *p* ≤ 0.000), and attraction (OR = 1.554; 95% CI 1.402–1.723; *p* ≤ 0.000) towards the advertisements with SM influencers/celebrities (Table 4).

## 4. Discussion

The present study primarily aimed to determine the relationship between using SM as a source of NI and suffering from EC. This study revealed that EC was prevalent among 50% of the young Saudi females who participated, which is a risk factor for ED development.

An important finding is that Instagram is the most used application for NI, which is supported by a prior study in which Instagram was more popular among females than males (*p* < 0.000) [45]. Instagram is the fastest growing social network globally [46] and is a type of SM that allows the sharing of content, such as images and videos, with brief text attached [47]. This could suggest that females prefer visual platforms because visual media are easier to remember than text. Previous studies suggest that frequent use of Instagram affects mental health, and it has been linked to symptoms of depression, anxiety, and low self-esteem [48,49]. Exposure to thin body ideals has a negative effect on mood and increases body dissatisfaction among females, which creates a breeding ground for DE behaviors [50,51,52]. This might explain why most participants who preferred Instagram suffered from HEC (69%).

However, these results do not corroborate the studies conducted by Basch et al. [53] and Quaidoo et al. [54] because the present study only focused on SM applications. It did not include other methods, such as traditional media, the internet, and health care providers, which were included in those studies and gained priority.

It was found that 7.2% of the participants searched for NI daily, and 20.7% of them had HEC. The authors believe that any erroneous internalization of nutrition and BI that is clearly promoted on SM, such as the diet industry, will increase the probability of repetitive searching for NI and the duration of searching. However, the findings were in contrast to the authors’ anticipation. Most participants who suffered from HEC were searching for NI monthly (48.3%) and spent almost 10–30 min (65.5%). A possible explanation could be that other sources of NI, outside the scope of SM, such as friends and family, who are relied upon as the primary source for information, were not included in this study. Regarding the duration of the search, there may be certain accounts on SM that searchers always resort to; therefore, they did not need more than 30 min to search.

This result differs from those reported previously by Onwe and Okocha [55], who found that 24.9% of the participants searched for nutritional/dietary information daily or every other day. This could be attributed to the type of information, and the current study focused on nutrition, unlike Onwe and Okocha [55], who explored many health topics.

Diets (51.7%) and healthy recipes (58.6%) were more nutritionally preferred among participants who suffered from HEC, which hinted at their intention to lose weight. There are several motivations for weight loss, including improved health, a perfect appearance, and compliance with new trends [56]. Furthermore, personal accounts of dietitians/nutritionists were the favorite source for NI among participants (71%), particularly those who suffered from HEC (82.8%). This reflects the findings of earlier research [57] showing that SM was an important source of information among young adults 18–30 years old, who mentioned that traditional sources of information seem to become out of date for modern orientation. Moreover, they stated that they could gain more information from others with a similar condition and that the search for information could contribute to avoiding doctors’ consultations.

Interestingly, participants who suffered from HEC were more interested in and interacted with advertising [AD] offered by SM celebrities/influencers than participants who suffered from MEC. This result mirrors that of a previous study that examined the effects of 50 accounts of influencers who promoted dieting, nutrition, and physical activity on the Instagram platform, which revealed that 84% of influencers, who were mainly female, tried to instill inaccurate concepts among their followers. These were concepts wherein happiness and beauty can be obtained by achieving clear BI through two main components, restricting food and exercising strictly [58].

In the present study, the prevalence of EC according to SCOFF was slightly higher at 41.3%, which concurs with a prior study in Palestine (38.3%) [59] and can be attributed to sociocultural factors that are common among Arab countries. A much higher value (48.8%) was reported in Vietnam, which might be due to an increase in the prevalence of underweight (45.3%) in the Vietnamese sample [60] compared to the current sample (14.2%).

The findings in the present study significantly differ from previous results reported in the literature [61,62,63,64]. This difference could be attributed to the early attention paid to the ED issue in European countries through research that contributed to the formation of recommendations that reflected on reducing the risk factors for suffering from ED.

The prevalence of EC according to ESP in this study was 44.3%, which is more than that reported in the USA (24%) by Hall et al. [65]. This may be due to the specifications of the current sample, which included a larger number (1092 versus 249), in addition to the age groups extending from 18 to 30 years of age. However, this finding was less than that reported in a newer American study (56%), which might be related to marital status, as most of the American sample was married 47% and 83.2% as compared to the current sample, which was mostly unmarried [66].

The last objective of this study was to determine the risk factors that predict suffering from EC. Searching for 31–60 min on SM for NI was correlated with suffering from EC (OR = 1.090; 95% CI 1.000–1.187; *p* ≤ 0.049); spending more than 30 min could be an indicator of being afraid of something, as previously hypothesized. Moreover, over-understanding the information might lead to negative effects on health when explicated incorrectly, and the way of offering information plays an essential role; when it comes to complex templates, it can create misconceptions, particularly among those unspecialized in nutrition.

Interest in diet topics was a predictor of EC (OR = 1.396; 95% CI 1.294–1.505; *p* ≤ 0.000). This was probably caused by BI dissatisfaction and the desire to engage in dieting to lose weight. This finding agrees with Fardouly et al. [67], who showed that there is an association between upward appearance comparisons to others who are more attractive (e.g., celebrity/model) on SM with less appearance satisfaction and more thoughts of dieting and exercising among females. Likewise, interest in dietary supplements (e.g., proteins, vitamins) was a predictor for EC (OR = 0.874; 95% CI 0.814–0.937; *p*≤ 0.000). This is perhaps a sign of the intention to use them as a meal replacement [68] in the belief that they can control calorie intake, leading to easy weight loss [69,70].

Surprisingly, personal accounts of dietitians/nutritionists were a predictive factor (OR = 1.170; 95% CI 1.071–1.277; *p* ≤ 0.001), highlighting the importance of checking educational background and ensuring registration and eligibility for providing information and counselling. The most interesting finding was that interaction with the content shown by SM celebrities/influencers, including new meals or restaurants AD or their diet experiments, was a more predictive factor for EC. This finding further supports previous research that reported that SM influencers had a significant effect on food purchasing behaviour among young adults [71]. Another study reported that influencers on Instagram had a clear impact on young adults regarding the choice of their meals. These influencers shared their food tasting experiences, thereby inspiring their curious followers to try the foods [72].

It seems that celebrities and influencers on SM get striking power over the behaviour of their followers, which can catalyse herd behaviour regarding diet and exercise when they are role models [73]. It was found that perceived health status was a predictive factor for EC; whenever perceived health was poor, a person was more likely to suffer from EC. This is in accordance with earlier research, which revealed that individuals with mental health disorders and unhealthy dietary behaviours were more likely to assess their health as poor (OR = 0.23; 95% CI, 0.10–0.56; *p* ≤ 0.001) [74]. Dieting and BMI were predictive factors for EC, which is consistent with the findings of Solmi et al. [61] that EC was correlated with obesity (OR = 2.1; 95% CI 1.3–3.4; *p* ≤ 0.002). As being overweight and obese are correlated with stigma (linked to weight), it is somewhat surprising that being underweight was a predictive factor for EC. Although by a small number (OR = 0.672), this finding is consistent with the study by Jach and Krystoń [75], which showed that most females encountered such stigmatization despite having a normal weight (according to BMI), as many believed their actual BI was larger than the ideal BI.

Finally, this study has several strengths. First, to our knowledge, this is the first study in Saudi Arabia to explore the relationship between using SM as a source of NI and suffering from EC. Second, this study used a validated questionnaire in which SCOFF and ESP achieved high sensitivity. Third, females in the age range of 18–30 years are considered to be at peak fertility, which means that all nutritional deficiencies will affect their reproductive health later. Fourth, the sample size was large (1092), which contributed to minimizing bias in data collection and increasing the accuracy of the prevalence measure. Fifth, using an online questionnaire was effective because it made the participants comfortable, and they did not feel ashamed to answer sensitive questions that assessed EC.

However, this study has some limitations that must be considered. First, the cross-sectional design explores the associations but does not determine the causes. Second, the convenience sample technique may have a bias, and the present study focused on three applications (Twitter, Instagram, and Snapchat), and, therefore, might not be generalized to all users of social media. Third, data were collected using self-reports from participants, which are likely to have a bias in weight and height. Fifth, the measures used [SCOFF and ESP] represent brief instruments that contribute to screening initial symptoms of those likely to have ED, but do not diagnose a specific type of ED.

## 5. Conclusions

The prevalence of EC was high among Saudi females; in particular, participants who were classified with HEC were from the Southern region. Instagram had more endorsement as a favorite source for NI, and the most time spent on the search ranged between 31–60 min. Diet and healthy recipes were the most interesting nutritional topics. Moreover, this study disclosed the risk factors for suffering from EC, most notably, interaction with and attraction to AD on SM, which is offered by influencers/celebrities regarding new meals or restaurants and their diet experiments.

The findings show that ED risks among young females are no longer a Western culture-specific issue, but have evolved into a worldwide social issue that requires more attention and research. Overall, SM could be a unique and promising platform for nutrition education, allowing easy access to the target audience. However, it still needs more caution before it can provide a useful and reliable source of information.

It should be noted that this study discovered that repeated searching for NI is an indicator for EC, which contradicts earlier literature that found that searching for information will play an effective role in increasing nutrition education and creating awareness of healthy eating and bodies, thereby decreasing the chance of developing concerns. Regardless, the authors firmly believe that their findings are the first of their kind, and there are many gaps that require further investigation.

## 6. Recommendations

(1)Creating clinics on SM platforms—specializing in nutrition targeting young females to provide nutritional counseling and enhance their healthy lifestyles—should be allowed as long as they are managed by government agencies.(2)Employing a registered dietitian/nutritionist through SM platforms to answer questions and provide advice would minimize the chance of dealing with delusive accounts or unreliable people.(3)Planning awareness campaigns intended to educate young females on how to deal with messages that spread through SM without any evidence regarding their truthfulness.(4)Prospective research should focus on individuals who are over 30 years of age, including males, with varied SM applications, such as WhatsApp and TikTok, utilizing more sensitive instruments to screen ED risks, such as the Eating Disorders Examination Questionnaire [EDE-Q 6.0] [76] and the Eating Disorders Diagnostic Scale [EDDS] [77].

## Figures and Tables

**Table 1 ijerph-19-16380-t001:** Socio-Demographic Characteristics and Anthropometrics of Participants.

Variables		All N = 1092	LEC N = 550 (50.3%)	MEC N = 513 (47%)	HEC N = 29 (2.7%)	*p*-Value
Age (*Mean* ± *SD*)		23.02 ± 3.47	23.01 ± 3.41	22.97 ± 3.53	24.03 ± 3.57	0.276 ^a^
Marital status (*n*, %)	Unmarried	909 (83.2%)	471 (85.6%)	413 (80.5%)	25 (86.2%)	0.074 ^b^
	Married	183 (16.8%)	79 (14.4%)	100 (19.5%)	4 (13.8%)	
Region of residence (*n*, %)	Eastern	248 (22.7%)	122 (22.2%)	122 (23.8%)	4 (13.8%)	0.451 ^b^
	Central	271 (24.8%)	139 (25.3%)	127 (24.8%)	5 (17.2%)	
	Western	222 (20.3%)	121 (22%)	95 (18.5%)	6 (20.7%)	
	Northern	187 (17.1%)	93 (16.9%)	88 (17.2%)	6 (20.7%)	
	Southern	164 (15%)	75 (13.6%)	81 (15.8%)	8 (27.6%)	
Perceived health status (*n*, %)	Good	657 (60.2%)	365 (66.4%)	278 (54,2%)	14 (48.3%)	0.000 **^b^
	Average	395 (36.2%)	174 (31.6%)	207 (40.4%)	14 (48.3%)	
	Poor	40 (3.7%)	11 (2%)	28 (5.5%)	1 (3.4%)	
Dieting (*n*, %)	Yes	176 (16.1%)	40(7.3%)	118 (23%)	18 (62.1%)	0.000 **^b^
	No	916 (83.9%)	510 (92.7%)	395 (77%)	11 (37.9%)	
Weight (kg) (*Mean* ± *SD*)		58.51 ±12.90	53.49 ± 10.49	63.56 ± 13.11	64.58 ± 13.73	0.000 **^a^
Hight (m) (*Mean* ± *SD*)		1.58 ± 0.06	1.58 ± 0.06	1.58 ± 0.06	1.59 ± 0.06	0.078 ^a^
BMI (kg/m^2^) (*Mean* ± *SD*)		23.38 ± 4.85	21.50 ± 3.96	25.29 ± 4.92	25.38 ± 5.14	0.000 **^a^
BMI categories (*n*, %)	Underweight (BMI < 18.5)	155 (14.2%)	126 (22.9%)	28 (5.5%)	1 (3.4%)	0.000 **^b^
	Normal (18.5 ≤ BMI ≤ 24.9)	614 (56.2%)	341 (62%)	258 (50.3%)	15 (51.7%)	
	Overweight (25 ≤ BMI ≤ 29.9)	206 (18.9%)	65 (11.8%)	132 (25.7%)	9 (31%)	
	Obesity (BMI ≥ 30)	117 (10.7%)	18 (3.3%)	95 (18.5%)	4 (13.8%)	

^a^ ANOVA test; ^b^ Chi-Square test; ** *p* ≤ 0.001; Abbreviations: *SD*, standard deviation; LEC, low eating concerns; MEC, moderate eating concerns; HEC, high eating concerns; BMI, body mass index; kg, kilogram; m, meter.

**Table 2 ijerph-19-16380-t002:** Prevalence of Eating Concerns.

Variables	All N = 1092	LEC N = 550 (50.3%)	MEC N = 513 (47%)	HEC N = 29 (2.7%)	*p*-Value
SCOFF					
SCOFF (*Mean* ± *SD*)	1.39 ± 1.14	0.61 ± 0.65	2.07 ± 0.91	3.93 ± 0.70	0.000 **^a^
SCOFF < 2 (*n*, %)		641 (58.7%)	SCOFF ≥ 2 (*n*, %)		451 (41.3%)
ESP					
ESP (*Mean* ± *SD*)	1.41 ± 1.04	0.76 ± 0.67	2.00 ± 0.89	3.31 ± 0.66	0.000 **^a^
ESP < 2 (*n*, %)		608 (55.7%)	ESP ≥ 2 (*n*, %)		484 (44.3%)
Total (SCOFF + ESP) (*Mean* ± *SD*)	2.80 ± 1.75	1.37 ± 0.69	4.07 ± 1.02	7.24 ± 0.58	0.000 **^a^

^a^ ANOVA test; ** *p* ≤ 0.001; Abbreviations: LEC, low eating concerns; MEC, moderate eating concerns; HEC, high eating concerns; SCOFF, sick, control, One stone, fat, food; ESP, eating disorders screen for primary care; *SD*, standard deviation.

**Table 3 ijerph-19-16380-t003:** Nutritional Information Search Behaviours.

Variables		All N = 1092	LEC N = 550 (50.3%)	MEC N = 513 (47%)	HEC N = 29 (2.7%)	*p*-Value
Favorite application for NI (*n*, %)	Instagram	587 (53.8%)	276 (50.2%)	291 (56.7%)	20 (69%)	0.069 ^b^
	Twitter	421 (38.6%)	229 (41.6%)	183 (35.7%)	9 (31%)	
	Snapchat	84 (7.7%)	45 (8.2%)	39 (7.6%)	0 (0%)	
Frequency of searching for NI (*n*, %)	Daily	79 (7.2%)	37 (6.7%)	36 (7%)	6 (20.7%)	0.033 *^b^
	Weekly	367 (33.6%)	174 (31.6%)	184 (35.9%)	9 (31%)	
	Monthly	646 (59.2%)	339 (61.6%)	293 (57.1%)	14 (48.3%)	
Number of minutes spent on search for NI (*n*, %)	10–30 min	841 (77%)	434 (78.9%)	388 (75.6%)	19 (65.5%)	0.160 ^b^
	31–60 min	205 (18.8%)	90 (16.4%)	106 (20.7%)	9 (31%)	
	Above 60 min	46 (4.2%)	26 (4.7%)	19 (3.7%)	1 (3.4%)	
Most interesting topics in NI ^c^ (*n*, %)	Diets	395 (36.2%)	128 (23.3%)	252 (49.1%)	15 (51.7%)	
	Dietary supplements	483 (44.2%)	274 (49.8%)	202 (39.4%)	7 (24.1%)	
	Nutritional advice for specific illness cases	208 (19%)	109 (19.8%)	92 (17.9%)	7 (24.1%)	
	General NI	638 (58.4%)	336 (61.1%)	288 (56.1%)	14 (48.3%)	
	Healthy recipes	611 (56%)	297 (54%)	297 (57.9%)	17 (58.6%)	
	Others	30 (2.7%)	22 (4%)	7 (1.4%)	1 (3.4%)	
Favorite source for NI on SM (*n*, %)	Governmental accounts	239 (21.9%)	148 (26.9%)	88 (17.2%)	3 (10.3%)	0.005 *^b^
	Dietitians’/Nutritionists’ accounts	775 (71%)	368 (66.9%)	383 (74.7%)	24 (82.8%)	
	Family and friends’ accounts	13 (1.2%)	5 (0.9%)	8 (1.6%)	0 (0%)	
	Celebrities’/Influencers’ accounts	65 (6%)	29 (5.3%)	34 (6.6%)	2 (6.9%)	
I cannot resist celebrities’ advertisements on SM about new meals or restaurants. Therefore, I am in a hurry to try them (*n*, %)	No	906 (83%)	477 (86.7%)	410 (79.9%)	19 (65.5%)	0.001 **^b^
	Yes	186 (17%)	73 (13.3%)	103 (20.1%)	10 (34.5%)	
I am attracted to SM celebrities’ experiences with specific diets and like the amazing results they achieve. Therefore, I apply it immediately without nutritional counselling (*n*, %)	No	953 (87.3%)	520 (94.5%)	418 (81.5%)	15 (51.7%)	0.000 **^b^
	Yes	139 (12.7%)	30 (5.5%)	95 (18.5%)	14 (48.3%)	

^b^ Chi-Square test; ^c^ Multiple Response; * *p* ≤ 0.05; ** *p* ≤ 0.001; Abbreviations: LEC, low eating concerns; MEC, moderate eating concerns; HEC, high eating concerns; NI. Nutritional information; SM, social media.

**Table 4 ijerph-19-16380-t004:** Risk Factors for Eating Concerns.

Variables		Odds Ratio (OR)	95% Confidence Interval (CI) for OR		*p*-Value ^d^
			Lower Bound	Upper Bound	
Number of minutes spent on search for NI	10–30 min	Reference Category			
	31–60 min	1.090	1.000	1.187	0.049 *
	>60 min	0.974	0.819	1.159	0.767
Interest in diets	No	Reference Category			
	Yes	1.396	1.294	1.505	0.000 **
Interest in dietary supplements	No	Reference Category			
	Yes	0.874	0.814	0.937	0.000 **
Interest in general NI	No	Reference Category			
	Yes	0.918	0.857	0.983	0.015 *
Favorite source for NI on SM	Governmental accounts	Reference Category			
	Dietitians’/Nutritionists’ accounts	1.170	1.071	1.277	0.001 **
	Family and friends’ accounts	1.325	0.977	1.796	0.070
	Celebrities’/Influencers’ accounts	1.126	0.959	1.323	0.146
I cannot resist celebrities’ advertisements on SM	No	Reference Category			
	Yes	1.170	1.072	1.277	0.000 **
I am attracted to SM celebrities’ experiences	No	Reference Category			
	Yes	1.554	1.402	1.723	0.000 **
Perceived health status	Good	Reference Category			
	Average	1.201	1.116	1.291	0.000 **
	Poor	1.335	1.122	1.589	0.001 **
Dieting	No	Reference Category			
	Yes	1.613	1.464	1.778	0.000 **
BMI categories	Normal	Reference Category			
	Underweight	0.672	0.587	0.770	0.000 **
	Overweight	1.330	1.210	1.462	
	Obesity	1.675	1.487	1.888	

^d^ Multinomial Logistic Regression; * *p* ≤ 0.05; ** *p* ≤ 0.001; Abbreviations: NI, Nutritional information; SM, social media; BMI, body mass index.

## Data Availability

The datasets used and analyzed during the current study are available from the corresponding authors on reasonable request.

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
