# Peer review of "Eating Concerns Associated with Nutritional Information Obtained from Social Media among Saudi Young Females: A Cross-Sectional Study"

_ijerph, 2022, doi:10.3390/ijerph192416380_

Round 1

Reviewer 1 Report

Dear authors, 

This is an interesting article regarding social media and eating disorders.

I believe that the introduction, methods, discussion and conclusion parts need a lot of modifications so that they meet the study's aims. There are basic information that could be omitted, paragraphs that need to be shortened, methodological gaps and conclusions that is not clear how are they drawn.

Kind regards

Author Response

Thank you very much for your comments and suggestions. We appreciate the time and effort that you and the reviewers have dedicated to providing your valuable feedback on my manuscript. We improved the methods section as: clarified the language of consultation regarding nutritional information from social networks in the materials and methods section, page 3, lines 130. Also, we add more information about the exclusion and inclusion criteria Page 3, lines 114–117.

In the discussion section we provide the essential interpretation based on our key findings. Also separated the recommendations from the conclusion and added them in a new section, page 11, line 371. Subsequently, the conclusion section became abbreviated

Reviewer 2 Report

Thank you very much for being able to participate in this review.

From my point of view I consider this subject highly novel, for which scientific evidence is still needed. The paper is well structured and correctly includes both the objectives and the methodology carried out. 

However, I think that some aspects that are not included in the current version should be pointed out:

1- Is the language of consultation of this information known from social networks, only English, English and Arabic, only Arabic, other languages?.

2- The content that is followed, the professionals, pages are only Arab or from other regions of the world?

Although the female population is more susceptible to these behaviors, it would also be an opportunity to know the impact of this on men.

Good job.

Author Response

Thank you very much for your comments and suggestions.

  • Is the language of consultation of this information known from social networks, only English, English and Arabic, only Arabic, other languages?

It was only focused on the Arabic content language.  Wherefore we added this information in the materials and methods section, page 3, lines 130.  

"This section discusses many elements, most notably, the favourite application for searching for NI, the preferred source for NI published in the Arabic language by Arab speakers, ……."

  • The content that is followed, the professionals, pages are only Arab or from other regions of the world?

It was only limited to Arab speakers. We added this information in the materials and methods section, page 3, line 130–131.

"This section discusses many elements, most notably, the favourite application for searching for NI, the preferred source for NI published in the Arabic language by Arab speakers, ……."

-Although the female population is more susceptible to these behaviors, it would also be an opportunity to know the impact of this on men.

Basically, we mentioned males as future research in the recommendations section, page 11, line 383.

"Prospective research should focus on individuals who are over 30 years of age, including males, …"

Reviewer 3 Report

Many thanks to the authors who worked hard on this article. I want to thank the MDPI editors who invited me to review this manuscript to estimate the prevalence of eating concerns among Saudi females, investigate the association between using social media as a source of nutritional information and eating concerns; and Determine the risk factors for suffering with eating concerns. The work would benefit from some improvements, which I describe below:

In my opinion, the section conclusion should be abbreviate

exclusion and inclusion criteria should be added to the methodology section 

Author Response

Thank you very much for your comments and suggestions.  

  • In my opinion, the section conclusion should be abbreviate

Accepted suggestion, therefore we separated the recommendations from the conclusion and added them in a new section, page 11, line 371. Subsequently, the conclusion section became abbreviated. 

  • exclusion and inclusion criteria should be added to the methodology section 

The inclusion criteria were included in the materials and methods section, Page 3, lines 114–117.

"Participation in this study was voluntary included the following inclusion criteria: Saudi females only, aged between 18–30 years, users of at least one of these applications (Twitter, Instagram, and Snapchat) to search for NI, non-pregnant, and non-breastfeeding, …….".

However, we accepted this suggestion and added the exclusion criteria (page 3, line 117).

"Participation in this study was voluntary included the following inclusion criteria: Saudi females only, aged between 18–30 years, users of at least one of these applications (Twitter, Instagram, and Snapchat) to search for NI, non-pregnant, and non-breastfeeding, whereas excluded those who did not have those criteria."
